# HHV-6B detection and host gene expression implicate HHV-6B as pulmonary pathogen after hematopoietic cell transplant

Joshua A. Hill [1,2,3] ✉, Yeon Joo Lee[4,5], Lisa K. Vande Vusse[1], Hu Xie[3], E. Lisa Chung [2], Alpana Waghmare [2,6], Guang-Shing Cheng[1,3], Haiying Zhu[7], Meei-Li Huang[7], Geoffrey R. Hill[3,8], Keith R. Jerome [2,7], Wendy M. Leisenring[3], Danielle M. Zerr[2,6], Sina A. Gharib[1], Sanjeet Dadwal[9,10] & Michael Boeckh[1,2,3,10]

Limited understanding of the immunopathogenesis of human herpesvirus 6B (HHV-6B) has prevented its acceptance as a pulmonary pathogen after hematopoietic cell transplant (HCT). In this prospective multicenter study of patients undergoing bronchoalveolar lavage (BAL) for pneumonia after allogeneic HCT, we test blood and BAL fluid (BALF) for HHV-6B DNA and mRNA transcripts associated with lytic infection and perform RNA-seq on paired blood. Among 116 participants, HHV-6B DNA is detected in 37% of BALs, 49% of which also have HHV-6B mRNA detection. We establish HHV-6B DNA viral load thresholds in BALF that are highly predictive of HHV-6B mRNA detection and associated with increased risk for overall mortality and death from respiratory failure. Participants with HHV-6B DNA in BALF exhibit distinct host gene expression signatures, notable for enriched interferon signaling pathways in participants clinically diagnosed with idiopathic pneumonia. These data implicate HHV-6B as a pulmonary pathogen after allogeneic HCT.

Lower respiratory tract disease (LRTD) remains a leading cause of morbidity and mortality after allogeneic hematopoietic cell transplant (HCT)[1-4]. Despite advances in pathogen-specific diagnostic methods, the causative pathogen is often elusive. Up to 10% of HCT recipients with LRTD are diagnosed with idiopathic pneumonia syndrome (IPS)[5], although current diagnostic approaches may miss potentially important pathogens[6]. Indeed, advances in diagnostic testing may have reduced the misclassification of infection-related LRTD as IPS, but IPS remains associated with high mortality[7,8].

Prior studies suggest that human herpesvirus 6B (HHV-6B) may be a pulmonary pathogen after allogeneic HCT[9-16]. HHV-6B is a DNA betaherpesvirus closely related to cytomegalovirus (CMV). HHV-6B infects most individuals during early childhood[17] and establishes latency in a broad range of cells, including bronchial glands[18,19]. HHV-6B DNA plasma detection (i.e., reactivation) occurs in ~40–60% of HCT recipients within six weeks after HCT[20,21]. Small studies of heterogeneous populations using varied methods to detect HHV-6B in blood or bronchoalveolar lavage fluid (BALF) have inconsistently implicated HHV-6B as a potential pulmonary pathogen[6,9–16]. We previously performed a large single-center retrospective study and detected HHV-6B DNA in BALF from 27% of 553 allogeneic HCT recipients with LRTD. Patients with HHV-6B DNA in BALF had significantly increased risk of

[1]Department of Medicine, University of Washington, 1959 NE Pacific St, Seattle, WA 98195, USA. [2]Vaccine and Infectious Disease Division, Fred Hutchinson Cancer Center, 1100 Fairview Ave N, Seattle, WA 98109, USA. [3]Clinical Research Division, Fred Hutchinson Cancer Center, 1100 Fairview Ave N, Seattle, WA 98109, USA. [4]Infectious Diseases Service, Department of Medicine, Memorial Sloan Kettering Cancer Center, 1275 York Ave, New York, NY 10065, USA. [5]Weill Cornell Medical College, 400 E 67th St, New York, NY 10065, USA. [6]Seattle Children's Hospital, 4800 Sand Point Way NE, Seattle, WA 98105, USA. [7]Department of Laboratory Medicine and Pathology, University of Washington, 1959 NE Pacific St, Seattle, WA 98195, USA. [8]Translational Science and Therapeutics Division, Fred Hutchinson Cancer Center, 1100 Fairview Ave N, Seattle, WA 98109, USA. [9]City of Hope National Medical Center, 1500 E Duarte Rd, Duarte, CA 91010, USA. [10]These authors contributed equally: Sanjeet Dadwal, Michael Boeckh. ✉e-mail: jahill3@fredhutch.org

death compared to subjects without HHV-6B detection, with and without co-pathogens[22]. However, testing for HHV-6B in BALF is not routinely performed, and there are no guidelines for HHV-6B testing in the setting of LRTD despite available therapies[2,23,24].

The diagnosis of pathogens causing pneumonia often relies on detecting their genomic DNA or RNA using PCR, but this approach has poor specificity for pathogens prone to colonization and/or asymptomatic shedding[25,26]. Establishing a causal link between HHV-6B and LRTD is further hindered by diagnostic methods that do not distinguish between viral shedding (i.e., genomic DNA detection in the absence of viral gene expression indicating latent infection) and lytic infection (i.e., genomic DNA plus viral gene expression indicating viral replication with cytopathic effect). For example, detection of messenger (m)RNA transcripts of CMV in BALF using reverse-transcription quantitative PCR (RT-qPCR) had greater specificity for CMV pneumonia than genomic DNA detection by qPCR[27]. The additive value of testing for viral gene transcription, the correlation of HHV-6 detection in blood and BALF, and whether our previous findings of an association between HHV-6B detection in BALF and mortality remain pertinent in contemporary patients are unknown.

Leveraging additional diagnostic approaches, such as genome-wide host expression profiling, may provide insight into the pathogen-specific immune pathways and further improve diagnostic strategies. Transcriptional profiling, an emerging tool for pathogen-associated disease assessment, can identify specific host signatures to discriminate between infectious and non-infectious causes of disease[28–35]. This approach has been used on blood and BALF to identify and discriminate between a broad range of pathogens in other contexts[33,36–42] and may be especially useful when the role of a pathogen is uncertain[29].

Few prospective studies have systematically preserved blood and BALF after allogeneic HCT for viral and host gene expression analyses. In this study, we prospectively enrolled adult allogeneic HCT recipients undergoing a BAL for evaluation of LRTD within 120 days of HCT from three centers and followed outcomes for up to 60 days after the BAL. We collected and tested paired blood and BALF samples with multiple molecular diagnostic techniques to comprehensively investigate the role of HHV-6B as a pulmonary pathogen after allogeneic HCT and determine the utility of whole blood gene expression signatures as a complement to pathogen-directed diagnostic testing.

## Results

### Participant characteristics and LRTD diagnoses

We prospectively enrolled 116 adult allogeneic HCT recipients undergoing 125 BAL events within 120 days after HCT at three institutions from July 2015 through December 2019 (Fig. 1). Plasma and BALF samples were not collected in one and three participants, respectively. Participant characteristics are described in Table 1 and were generally similar between those with and without HHV-6B detection in BALF. Of note, a lower proportion of participants with higher HHV-6B viral load were receiving antiviral therapy at the time of sample collection. The median number of days between hematopoietic cell infusion and BAL was 37 (interquartile range [IQR], 19–76).

We defined LRTD as any abnormal radiographic finding with lower respiratory tract symptoms (see Online Methods). Bacterial, fungal, and viral causes were determined according to consensus guidelines[2,43–45]. Idiopathic pneumonia syndrome (IPS) was defined according to American Thoracic Society guidelines, requiring that no infection be suspected or detected[2]. If a microbiologic or non-infectious (i.e., IPS) cause of LRTD was not determined based on a standardized review of microbiologic, histopathologic, and other diagnostic criteria, the event was categorized as 'other'. LRTD was most often due to multifactorial causes followed by invasive fungal pneumonia. Specific causes of LRTD are detailed in Table S1.

### HHV-6B DNA detection was frequent in BALF and less common in paired plasma samples

HHV-6B DNA was detected in 45 of 122 BALF samples (37%) (Fig. 1). In comparison, HHV-6B DNA was detected in 19 of 124 plasma samples (15%). Plasma detection of HHV-6B DNA had poor sensitivity (29.3%; 95% CI, 15.3–43.2%) but high specificity (94.4%; 95% CI, 89–99.7%) for BALF detection of HHV-6B DNA. Plasma and BALF viral load were moderately correlated (Figure S1). The distribution of BALF HHV-6B DNA and mRNA detection over time and by LRTD category is depicted in Fig. 2A, B. The proportion of BALs with HHV-6B DNA detection varied over time in a non-specific pattern. Participants with multiple causes of LRTD had the highest proportion of BALF HHV-6B detection (58%), and those with 'other' non-specific causes of LRTD had the lowest proportion of positivity (18%). The median BALF HHV-6B DNA viral load was 2.6 log$_{10}$ copies/mL (IQR, 2.0–3.4) and was highest in participants tested in the first 60 days after HCT (Fig. 2C). Participants with 'other' non-specific causes of LRTD had the lowest viral loads (Fig. 2D). We did not detect HHV-6 species A in any samples.

### Higher HHV-6B DNA viral load predicted HHV-6B mRNA detection in BALF

Among the 45 BALF and 19 plasma samples with HHV-6B DNA detected, HHV-6B mRNA transcripts for either *U38* or *U90* were detected in 22 (49%) BALF and 16 (84%) whole blood samples; both transcripts were detected in 10 (22%) BALF and 12 (63%) whole blood samples (Table S2). HHV-6B mRNA detection over time and by LRTD category had similar patterns as observed with HHV-6B DNA (Fig. 2A, B; Figure S2). HHV-6B mRNA detection tended to occur in BALF obtained earlier after HCT (median, 21 days; IQR, 18–26 compared to HHV-6B negative BALF: median, 43 days; IQR, 20–81).

We next aimed to examine the ability of HHV-6B viral load to distinguish between viral replication, a state during which mRNA is expressed, from viral shedding, during which mRNA expression is expected to be minimal or absent. We constructed ROC curves to measure the performance of BALF HHV-6B DNA viral loads for predicting HHV-6B mRNA expression. This demonstrated that HHV-6B DNA ≥ 2.3 log$_{10}$ copies/mL (218 copies/mL) in BALF had a high area under the curve (AUC) of 0.93 (95% CI, 0.89–0.98) for predicting detection of HHV-6B mRNA expression (*U38* or *U90*; Figure S3A) with a sensitivity, specificity, positive predictive value, and negative predictive value of 85%, 88%, 61%, and 96%, respectively (Table S3). A BALF viral load threshold of ≥2.8 log$_{10}$ copies/mL (578 copies/mL) in BALF performed the best for predicting detection of both HHV-6B mRNA transcripts (Figure S3B).

### BALF HHV-6B DNA detection at levels that predict mRNA detection is associated with increased overall mortality and pulmonary death

Among 113 participants with a tested BALF sample, 35 (31%) died for any reason and 27 (24%) died from respiratory failure within 60 days of the BAL. Participants with HHV-6B DNA detection at or above the threshold that predicts mRNA expression (≥2.3 log$_{10}$ copies/ml) in BALF (*n* = 27) had increased risk for overall mortality (adjusted hazard ratio [aHR], 1.81; 95% CI, 0.89–3.67; *p* = 0.099) and death from respiratory failure (aHR, 2.35; 95% CI, 1.08–5.11; *p* = 0.032) in models adjusted for age, maximum supplemental oxygen use (within 24 h prior to the BAL), and maximum corticosteroid dose (within 14 days prior to the BAL) (Fig. 3A, B, Figure S4, Table S4). When stratified by a higher BALF HHV-6B DNA viral load threshold of ≥2.8 log$_{10}$ copies/mL that corresponded to detection of both mRNA transcripts, there was a stronger association with overall mortality (aHR, 2.86; 95% CI, 1.24–6.60; *p* = 0.014) and death from respiratory failure (aHR, 2.79; 95% CI, 1.11–7.03; *p* = 0.03) (Fig. 3C, D, Table S5). There was no significant difference in overall mortality or pulmonary death between underlying LRTD diagnoses, and inclusion of this variable in adjusted

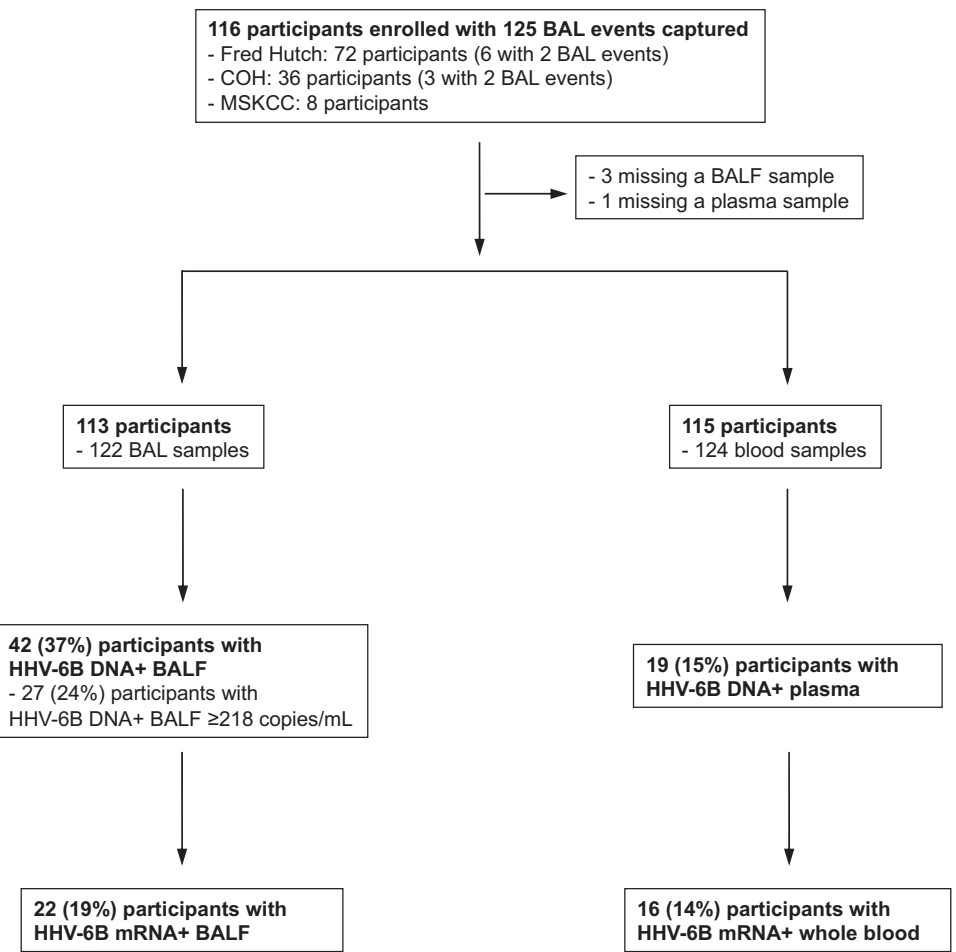

**Fig. 1 | Consort diagram.** Number of enrolled participants, samples, and HHV-6B test results.

models demonstrated a similar effect size of HHV-6B on outcomes at the 2.3 $\log_{10}$ copies/ml threshold but an even higher effect size at the 2.8 $\log_{10}$ copies/ml threshold (Tables S4 and S5). In addition, inclusion of a variable for use of antiviral therapy at the time of sample collection resulted in a similar increase in the strength of the association of HHV-6B detection with overall mortality and death from respiratory failure at the higher 2.8 $\log_{10}$ copies/ml threshold (Tables S4 and S5).

**Genome-wide host expression profiling distinguishes between participants with and without HHV-6B detection in BALF**
We identified a subgroup of 54 participants who had a well-documented single cause of LRTD based on clinical testing (bacterial, $n = 7$; viral, $n = 17$; fungal, $n = 17$; IPS, $n = 13$) and a whole blood collection with sufficient quantity and quality of extracted RNA for bulk sequencing. One participant did not have a BALF sample available but is included in analyses. Characteristics of this cohort were similar to the overall cohort (Table S6). Key biologic variables of interest are stratified by LRTD category and HHV-6B detection in Table S7. HHV-6B ≥ 218 copies/mL in BALF was detected in 10 of 53 (19%) individuals in this subgroup; any level of plasma HHV-6B DNA was detected in 8 of 54 (15%) individuals. For gene expression comparisons, a BALF sample was considered positive for HHV-6B detection at levels ≥2.3 $\log_{10}$ copies/mL. Of note, 19 of the 54 patients (35%) were receiving antiviral therapy with ganciclovir, foscarnet, or cidofovir for CMV or adenovirus at the time of sample collection for this study. Notably, only two of these patients had HHV-6B in the BALF and/or the blood, so most patients with viremia from another virus were in the 'no HHV-6B' comparator group (Table S7). Only one participant in the bacterial

subgroup had HHV-6B detection in BALF or blood, so this subgroup was not further analyzed. Because we measured transcriptional signals in the blood, we categorized individuals who had HHV-6B detection in plasma in the same group as those with BALF detection. A sensitivity analysis defining participants based solely on BALF detection of HHV-6B did not substantially alter our findings but slightly reduced power for detecting differentially expressed genes and pathways.

We identified >10,000 genes per comparison with sufficient expression count for analysis. Multidimensional scaling using principal components analysis (PCA) of the entire transcriptome suggested segregation between patients with viral LRTD who also had HHV-6B detection in BALF or plasma and those without HHV-6B detection; similar segregation by HHV-6B detection was evident in the IPS subgroup but not in the fungal subgroup (Fig. 4A–C). This observation indicates that detection of HHV-6B was associated with a stronger transcriptional signal in the viral and IPS subgroups compared to the fungal subgroup. Patients with HHV-6B viremia appeared to more closely segregate with those who had HHV-6B detected in BALF, although comparisons were limited by the number of events. These findings indicate that in subjects with LRTD from another virus or LRTD without a clinically identified infectious cause (i.e., IPS), detection of HHV-6B DNA is associated with widespread differences in the host transcriptional response.

We used DESeq2's statistical approach to identify genes differentially expressed between subjects with and without HHV-6B detection within each LRTD group. Using a combined fold-change > +/−1.5 and an FDR < 5% threshold, we found 929 differentially regulated genes in subjects with versus without HHV-6B detection (434 up and

## Table 1 | Demographic and clinical variables at the time of the BAL

| | Negative HHV-6B DNA in BAL (N = 71) | Positive HHV6-B DNA in BAL (any level) (N = 42)[a] | Positive HHV6-B DNA in BAL ≥ 2.3 log10 copies/mL (N = 27)[a] | Total (N = 113[b]) |
|---|---|---|---|---|
| **Age, years** | | | | |
| <=20 | 0 (0%) | 1 (2%) | 1 (4%) | 1 (1%) |
| 21–40 | 17 (24%) | 15 (36%) | 10 (37%) | 32 (28%) |
| 41–60 | 27 (38%) | 10 (24%) | 6 (22%) | 37 (33%) |
| >60 | 27 (38%) | 16 (38%) | 10 (37%) | 43 (38%) |
| **Female sex** | 27 (38%) | 22 (52%) | 13 (48%) | 49 (43%) |
| **Race** | | | | |
| Caucasian | 53 (75%) | 33 (79%) | 20 (74%) | 86 (76%) |
| Non-Caucasian | 15 (21%) | 9 (21%) | 7 (26%) | 24 (21%) |
| Unknown | 3 (4%) | 0 (0%) | 0 (0%) | 3 (3%) |
| **Year of HCT** | | | | |
| 2015 | 6 (8%) | 0 (0%) | 0 (0%) | 6 (5%) |
| 2016 | 12 (17%) | 5 (12%) | 4 (15%) | 17 (15%) |
| 2017 | 22 (31%) | 16 (38%) | 11 (41%) | 38 (34%) |
| 2018 | 18 (25%) | 7 (17%) | 5 (19%) | 25 (22%) |
| 2019 | 13 (18%) | 14 (33%) | 7 (26%) | 27 (24%) |
| **Center** | | | | |
| City of Hope | 20 (28%) | 16 (38%) | 9 (33%) | 36 (32%) |
| Fred Hutch | 46 (65%) | 24 (57%) | 17 (63%) | 70 (62%) |
| MSKCC | 5 (7%) | 2 (5%) | 1 (4%) | 7 (6%) |
| **CMV serostatus** | | | | |
| D- and R- | 10 (14%) | 4 (10%) | 3 (11%) | 14 (12%) |
| D+ or R+ | 61 (86%) | 36 (86%) | 22 (81%) | 97 (86%) |
| Missing | 0 (0%) | 2 (5%) | 2 (7%) | 2 (2%) |
| **HCT comorbidity index score[c]** | | | | |
| 0 (low) | 8 (11%) | 3 (7%) | 3 (11%) | 11 (10%) |
| 1–2 (intermediate) | 20 (28%) | 11 (26%) | 8 (30%) | 31 (27%) |
| >=3 (high) | 43 (61%) | 28 (67%) | 16 (59%) | 71 (63%) |
| **HLA D/R status** | | | | |
| Matched related | 9 (13%) | 7 (17%) | 6 (22%) | 16 (14%) |
| Matched unrelated | 26 (37%) | 14 (33%) | 6 (22%) | 40 (35%) |
| Mismatched related | 11 (15%) | 8 (19%) | 7 (26%) | 19 (17%) |
| Mismatched unrelated | 24 (34%) | 12 (29%) | 7 (26%) | 36 (32%) |
| Missing | 1 (1%) | 1 (2%) | 1 (4%) | 2 (2%) |
| **Donor cell source** | | | | |
| Peripheral blood | 51 (72%) | 33 (79%) | 18 (67%) | 84 (74%) |
| Bone marrow | 16 (23%) | 7 (17%) | 7 (26%) | 23 (20%) |
| Umbilical cord blood | 4 (6%) | 2 (5%) | 2 (7%) | 6 (5%) |
| **Myeloablative conditioning[d]** | 32 (45%) | 14 (33%) | 10 (37%) | 46 (41%) |
| **Maximum corticosteroid use pre-BAL[e]** | | | | |
| None | 39 (55%) | 17 (40%) | 11 (41%) | 56 (50%) |
| <1 mg/kg/day | 13 (18%) | 18 (43%) | 12 (44%) | 31 (27%) |
| ≥1 mg/kg/day | 19 (27%) | 7 (17%) | 4 (15%) | 26 (23%) |
| **Maximum oxygen use pre-BAL > 2 L/min[f]** | 30 (42%) | 21 (50%) | 14 (52%) | 51 (45%) |

## Table 1 (continued) | Demographic and clinical variables at the time of the BAL

| | Negative HHV-6B DNA in BAL (N = 71) | Positive HHV6-B DNA in BAL (any level) (N = 42)[a] | Positive HHV6-B DNA in BAL ≥ 2.3 log10 copies/mL (N = 27)[a] | Total (N = 113[b]) |
|---|---|---|---|---|
| **WBC count pre-BAL[g]** | | | | |
| >1000 cells/mm³ | 50 (70%) | 31 (74%) | 7 (26%) | 81 (72%) |
| ≤1000 cells/mm³ | 17 (24%) | 10 (24%) | 19 (70%) | 27 (24%) |
| Missing | 4 (6%) | 1 (2%) | 1 (4%) | 5 (4%) |
| **ALC pre-BAL[g]** | | | | |
| >300 cells/mm³ | 27 (38%) | 11 (26%) | 17 (63%) | 38 (34%) |
| ≤300 cells/mm³ | 33 (46%) | 26 (62%) | 7 (26%) | 59 (52%) |
| Missing | 11 (15%) | 5 (12%) | 3 (11%) | 16 (14%) |
| **ANC pre-BAL[g]** | | | | |
| >500 cells/mm³ | 48 (68%) | 29 (69%) | 17 (63%) | 77 (68%) |
| ≤500 cells/mm³ | 12 (17%) | 8 (19%) | 7 (26%) | 20 (18%) |
| Missing | 11 (15%) | 5 (12%) | 3 (11%) | 16 (14%) |
| **Day of BAL post-HCT (median, IQR)** | 37 (15-68) | 42 (20–79) | 31 (20–72) | 37 (19–76) |
| **Antiviral therapy at time of sample collection[h]** | 24 (34%) | 12 (29%) | 6 (22%) | 36 (32%) |
| **LRTD cause[i]** | | | | |
| Bacterial | 7 (10%) | 3 (7%) | 2 (7%) | 10 (9%) |
| Viral | 12 (17%) | 6 (14%) | 3 (11%) | 18 (16%) |
| Fungal | 16 (23%) | 10 (24%) | 8 (30%) | 26 (23%) |
| IPS | 14 (20%) | 4 (10%) | 4 (15%) | 18 (16%) |
| Other | 9 (13%) | 2 (5%) | 0 (0%) | 11 (10%) |
| Multifactorial | 13 (18%) | 17 (40%) | 10 (37%) | 30 (27%) |

Data are presented as number (percentage), unless otherwise indicated.

*HHV-6B* human herpesvirus 6B, *BAL* bronchoalveolar lavage, *D* donor, *R* recipient, *HLA* human leukocyte antigen, *D* donor, *R* recipient, *WBC* white blood cell, *ALC* absolute lymphocyte count, *ANC* absolute neutrophil count, *LRTD* lower respiratory tract disease, *IPS* idiopathic pneumonia syndrome.

[a]Groups are not mutually exclusive.
[b]3 of the 116 enrolled participants did not have a BAL fluid sample available for testing and are excluded from this Table and from the analyses that incorporate BAL fluid HHV-6B test results.
[c]Based on the HCT-comorbidity index.
[d]Myeloablative regimens included any regimen containing ≥800 cGY TBI, any regimen containing carmustine/etoposide/cytarabine/melphalan (BEAM), or any regimen containing busulfan/cyclophosphamide with or without antithymocyte globulin.
[e]Within 14 days pre-BAL, based on prednisone-equivalent dosing.
[f]Within 24 h preceding the BAL.
[g]Closest sample within 3 days pre-BAL.
[h]Ganciclovir, foscarnet, or cidofovir for CMV or adenovirus.
[i]Additional details on specific causes are in Table S1.

495 down) in the viral LRTD subgroup, 51 differentially expressed genes (46 up and 5 down) in the IPS subgroup, but none differentially expressed among patients with fungal LRTD.

To further examine potential gene expression differences, we performed pathway analysis using Gene Set Enrichment Analysis (GSEA) to define distinct groups of genes that share common biological function, chromosomal location, or regulation in each LRTD category. In participants with a clinically determined viral cause of LRTD, additional detection of HHV-6B was associated with upregulation of pathways involved in mitochondrial function/oxidative phosphorylation, oxidative stress, transcription and translation, cell cycle, viral infection, and several metabolic processes (Fig. 4D). Intriguingly, downregulated processes were highly enriched in immune responses, particularly T-cell receptor signaling and various interleukin-mediated pathways. In the IPS subgroup, participants with detection of HHV-6B

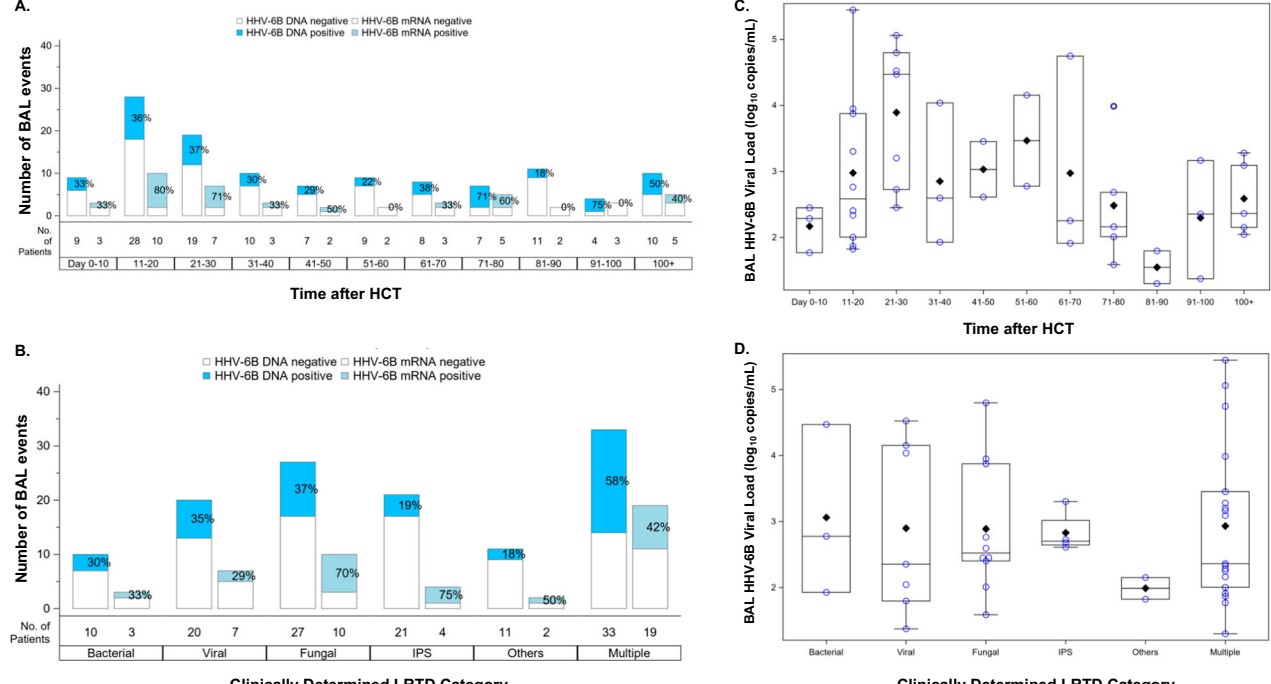

**Fig. 2 | The distribution of BALF HHV-6B DNA and mRNA detection over time and by LRTD category.** Panels (**A**) and (**B**) show data from 122 BAL events in 113 participants. The first bar in each time period or category indicates the proportion of BALs with HHV-6B DNA detection; the second bar indicates the proportion with HHV-6B mRNA detection, among those with HHV-6B DNA detection (samples without HHV-6B DNA detection were not tested for mRNA). Panels (**C**) and (**D**) show data from 45 BAL events (in 42 participants) that had HHV-6B DNA detection. The boxes represent the interquartile range, the horizontal lines and diamonds within the boxes represent the median and mean, respectively, and the upper and lower whiskers extend to the third and first quartiles plus or minus 1.5 times the interquartile range, respectively. Circles represent data points. LRTD indicates lower respiratory tract disease.

had highly enriched interferon signaling pathways, as well as pathways associated with allograft rejection, oxidative phosphorylation, viral infection, cell cycle, transcription, and translation (Fig. 4E). Down-regulated gene sets included Rho GTPase and Notch signaling, extracellular matrix/cellular remodeling and some immune-related pathways. In the fungal subgroup, participants with HHV-6B detection had widespread upregulation of immune-related pathways, with the most significant mapping to interferon signaling as well as numerous other immuno-inflammatory programs (Fig. 4F).

## Discussion

In this prospective, multicenter study of allogeneic HCT recipients with LRTD within 120 days of HCT, we demonstrate frequent detection of HHV-6B DNA and mRNA transcripts in BALF samples. HHV-6B was detected as a co-pathogen among individuals with other infectious causes of pneumonia and as the only pathogen in individuals thought to have an idiopathic non-infectious cause of pneumonia. Using HHV-6B DNA viral load thresholds identified to be predictive of HHV-6B mRNA transcription as a surrogate for active viral infection, we found that HHV-6B detection in BALF was independently associated with an increased risk for both overall mortality and death due to respiratory failure. Furthermore, genome-wide host gene expression profiling demonstrated distinct transcriptional signatures among participants with versus without BALF HHV-6B detection. Collectively, these results imply meaningful biological consequences when HHV-6B is detected in BALF from allogeneic HCT recipients with LRTD and support HHV-6B as a distinct pulmonary pathogen in this clinical context.

HHV-6B is an important but incompletely understood pathogen in the setting of allogeneic HCT. It is the most frequent cause of infectious encephalitis[46], but its role in other disease states is less well defined. Given its similarity to the closely related betaherpesvirus of CMV, a major cause of post-HCT pneumonitis prior to the

implementation of diagnostic and preventive strategies[47], there is a high likelihood that HHV-6B has similar pathogenicity. Indeed, in a unique study using a HCT mouse model infected with murine roseolovirus (a homolog of HHV-6), investigators demonstrated viral reactivation in the lung that induced an IPS-like phenotype[48]. These findings are concordant with a study demonstrating HHV-6 as the most frequently detected pathogen in individuals diagnosed with IPS after allogeneic HCT, who had worse outcomes than those without occult pathogen detection[6]. Identifying a potential role of HHV-6B in causing or contributing to LRTD is of significant clinical interest given the ready availability of diagnostic tests and antiviral therapies.

In the current study, we found that 37% of BAL events had detectable HHV-6B DNA. The observation of generally higher viral loads and more frequent detection of HHV-6B in BALs obtained earlier after HCT are consistent with the epidemiology of HHV-6B reactivation and HHV-6B-encephalitis after HCT[21]. Detection of HHV-6B in BALF in the absence of concurrent detection in the plasma also points to independent, compartmentalized reactivation of HHV-6B within the lung parenchyma.

A unique strength of this study was prospective collection and processing of BALF to allow testing for mRNA gene transcripts. We used targeted RT-qPCR to test for two HHV-6B-specific transcripts, *U38* (a late DNA polymerase gene) and *U90* (an immediate early gene), which have high sensitivity and specificity for viral replication in the context of lytic infection and would be unlikely to be detected in the context of an inactive, latent infection[49–53]. We found that HHV-6B DNA viral loads ≥2.3 and ≥2.8 log₁₀ copies/mL were highly predictive of detecting one or both mRNA transcripts, respectively. This evidence suggests that these thresholds may distinguish between viral shedding and lytic infection to reduce the 'signal-to-noise' ratio when testing cellular samples like BALF. Strikingly, these values are very similar to the HHV-6B DNA viral load threshold of 2.5 log₁₀ copies/mL identified

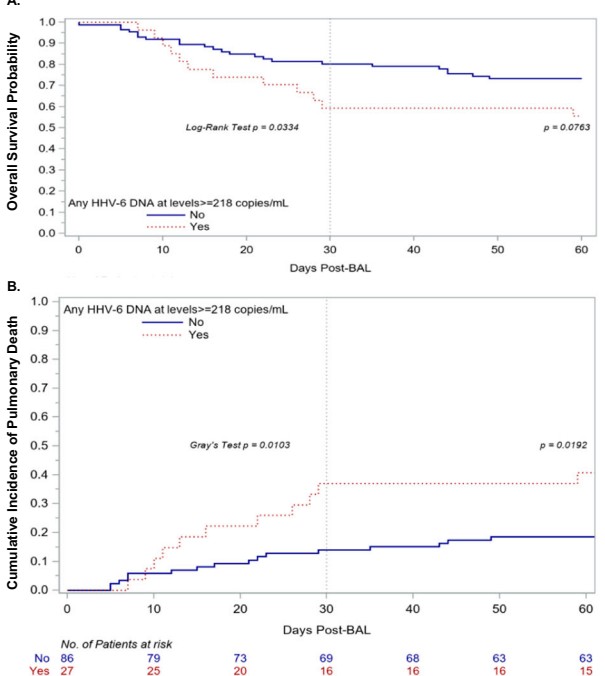

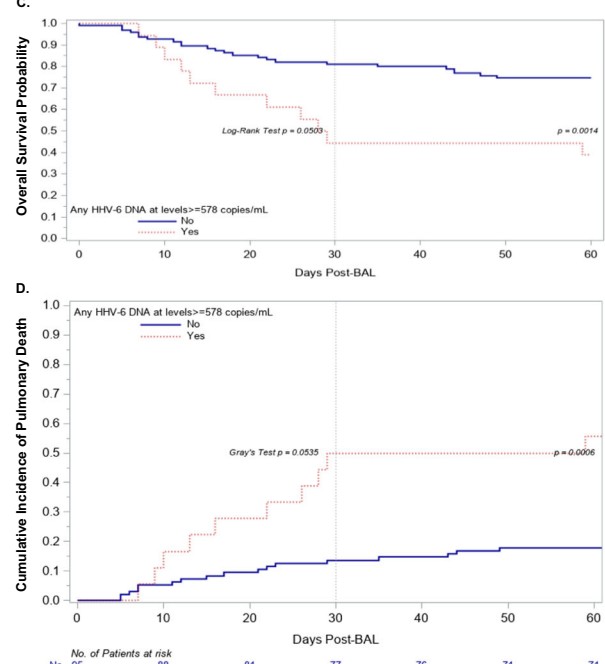

**Fig. 3 | Kaplan–Meier and cumulative incidence plots of time to overall mortality and death due to respiratory failure, respectively, among patients with and without HHV-6B detection in the BAL at different viral load thresholds.** Participants with HHV-6B DNA ≥ 218 copies/ml (2.3 log$_{10}$ copies/mL) in BALF based on the threshold predictive of detection of at least one of two mRNA transcripts ($n$ = 27) had increased overall mortality (**A**) and death from respiratory failure (**B**). Participants with HHV-6B DNA ≥ 578 copies/ml (2.8 log$_{10}$ copies/mL) in BALF based

on the threshold predictive of detection of both mRNA transcripts ($n$ = 18) had a greater increased risk for overall mortality (**C**) and death from respiratory failure (**D**). Deaths due to other causes were considered competing events in the cumulative incidence curves. The first and second log-rank or Gray's test statistics in the figures depict the unadjusted comparison of the curves at day 30 and day 60 post-BAL, respectively, and were two-sided. Results from adjusted Cox regression models are detailed in Tables S4–S5.

in our aforementioned retrospective study, in which viral loads above this level were associated with mortality and death from respiratory failure with effect sizes similar to the current study[22]. There was a suggestion that HHV-6B viral loads were lower in those receiving antivirals targeting related viruses such as CMV and adenovirus, which is not surprising given that these drugs have activity against HHV-6B[24]. This was similar to findings of our prior retrospective study[22]. Although this could impact our findings, one would expect this to bias the HHV-6B negative group to having worse outcomes, as requiring treatment for CMV or adenovirus is associated with lower survival[54]. In adjusted models that accounted for receipt of antiviral therapy, we observed a similar effect size of HHV-6B on outcomes at the 2.3 log$_{10}$ copies/ml threshold but an even higher effect size at the 2.8 log$_{10}$ copies/ml threshold for both overall mortality and pulmonary death. Similar findings were demonstrated in models incorporating LRTD subgroups.

Further evidence of tissue-invasive disease was demonstrated in our prior study using RNA in situ hybridization to detect HHV-6 gene expression in lung tissue from three patients with HHV-6B DNA in the BALF. Similar findings have been demonstrated for CMV in a study of allogeneic HCT recipients, wherein BALF CMV > 2.7 log$_{10}$ copies/mL (500 copies/mL) distinguished between CMV pneumonia and viral shedding without CMV pneumonia[26]. In addition, that study demonstrated a distribution of CMV detection across LRTD categories similar to that observed for HHV-6B in the current analysis[26].

Host gene expression profiling is increasingly reported as a highly discriminatory technique for pathogen-associated disease assessment[30]. Pathogens trigger specific host transcriptional programs by interacting with pattern recognition receptors on leukocytes and other cells to mediate immune responses. Thus, unique transcriptional signatures may be defined to discriminate causes of LRTD, and this approach has been used with blood and BALF for a range of pathogens

in other clinical contexts[33,36–42]. One study identified primary HHV-6 infection in healthy children as a driver of interferon and toll-like receptor signaling[29], but studies in HCT recipients are lacking.

In this study, we identified differential up- and down-regulation of whole blood gene expression pathways among patients with versus without HHV-6B detection in BALF. Notably, HHV-6B detection in participants with clinically diagnosed IPS and fungal LRTD was associated with a strong host interferon response and enrichment in a variety of pathways consistent with active viral infection, implicating HHV-6B as an independent but unrecognized cause of pneumonitis. We did not observe similar enrichment in interferon pathways in the viral infection subgroup, but this was anticipated given that interferon responses are expected to be elevated due to both HHV-6 viral infection as well as non-HHV-6 viral infections, so there was no differential expression relevant to each infection type; rather, differences in other unique pathways were observed. We also note 19 of the 54 patients (35%) in the transcriptomic analyses were receiving antiviral therapy with ganciclovir, foscarnet, or cidofovir at the time of sample collection for this study. Notably, only 1 had HHV-6B in the BALF and the blood, and 1 had HHV-6B in the blood only. Thus, although the use of antivirals was not equally distributed between comparison groups, almost all events were in the 'No HHV-6B' comparator group. This is reassuring that viremia due to other viruses was unlikely to be driving the observed differences the IPS and fungal groups with HHV-6B detection. The observation that most detection of HHV-6B was in the group not receiving antivirals is concordant with the known activity of these therapies for suppressing HHV-6B reactivation and infection. Further support for the role of HHV-6B detection in the lungs as an active and distinct infection in participants clinically diagnosed with IPS was our observation through PCA that samples from patients with HHV-6B detection in BALF (but not plasma) clustered with samples

**A. Viral LRTD**

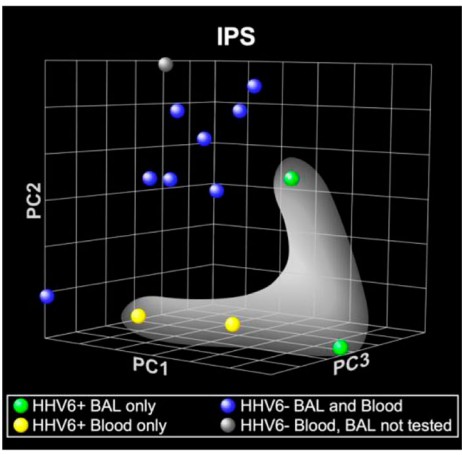

**D. Viral LRTD**

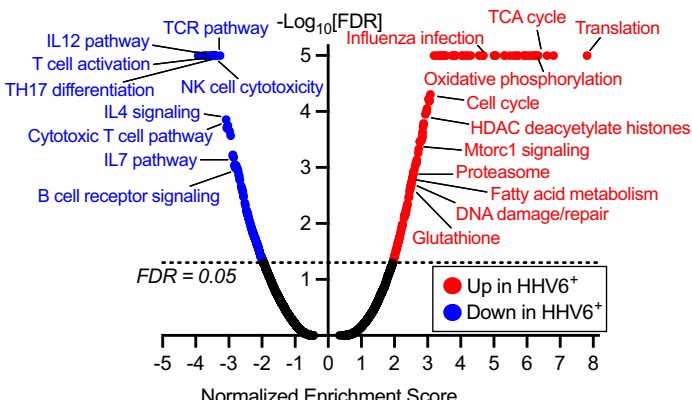

**B. IPS**

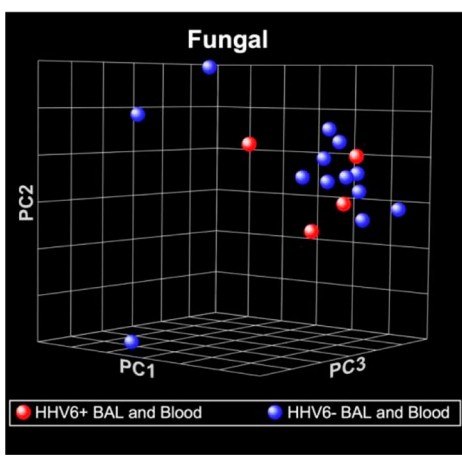

**E. IPS**

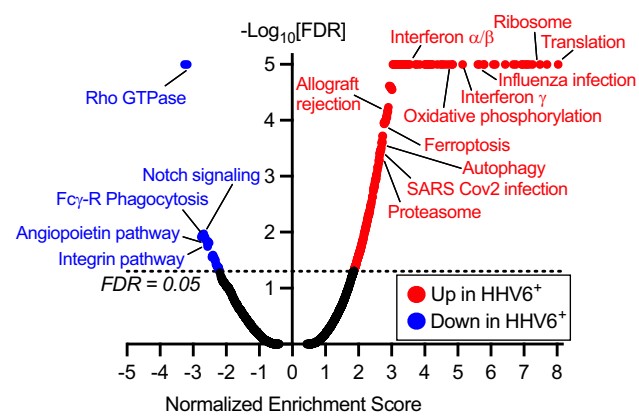

**C. Fungal LRTD**

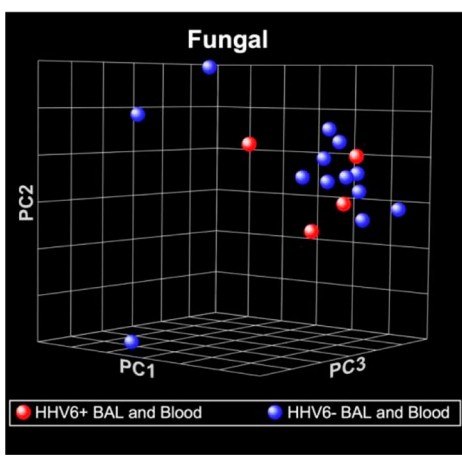

**F. Fungal LRTD**

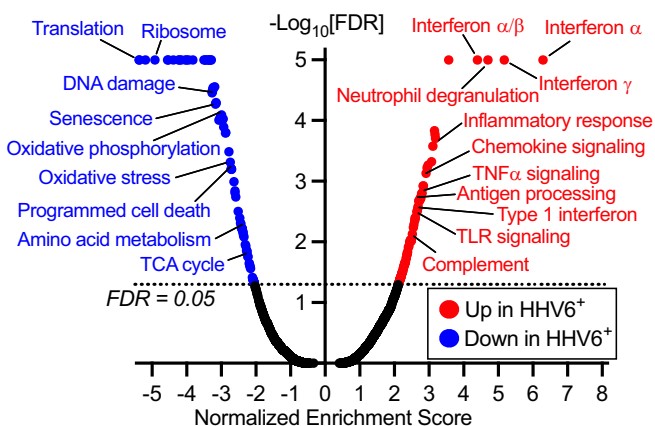

from patients who had only plasma detection—and both groups segregated from HHV-6B negative samples.

Strengths of this study include prospective enrollment and collection of paired BALF and plasma samples from a large cohort of contemporary HCT recipients at three centers. We preserved BALF for RNA analysis to examine key insights into the correlation between HHV-6B DNA detection and viral transcription in HCT patients with LRTD. Because isolation and testing of mRNA is labor-intensive, not

routinely available, and currently infeasible in a clinical setting, identification of a relevant genomic DNA threshold (for which testing is routinely available) is critical for clinical translation. Paired whole blood samples used for gene expression profiling generated additional observations in support of HHV-6B as a pulmonary pathogen and pave the way for similar approaches to disease definition in the HCT population. Although the transcriptional signal associated with HHV-6B detection was weaker in the fungal LRTD subgroup, GSEA's

**Fig. 4 | Genome-wide host expression profiling distinguishes between patients with and without HHV-6B detection.** Panels (**A**–**C**) depict multidimensional scaling using principal components analysis (PCA) of the entire transcriptome from whole blood to compare between participants with and without HHV-6B detection, among those with a specific clinically determined LRTD diagnosis. Among 17 participants with a viral pneumonia, PCA analysis suggested segregation between those with HHV-6B detection in the BAL and/or plasma (white highlight) versus those without (**A**). Among 13 participants with IPS, PCA analysis also suggested segregation between those with HHV-6B detection in the BAL and/or plasma (white highlight) versus those without (**B**). No clear pattern was seen among 17

participants with fungal pneumonia (**C**). Panels (**D**–**F**) display volcano plots of results from pathway analysis using Gene Set Enrichment Analysis (GSEA) to define and compare transcriptional programs by HHV-6B detection status in the same subgroups as panels (**A**–**C**). Each plot depicts gene sets that are either significantly upregulated (to the right of the plot) or downregulated (to the left of the plot) in participants with HHV-6B detection in BALF and/or plasma compared to those without. A false discovery rate (FDR) < 0.05 was used to designate significant enrichment of gene sets. Note: Only one patient in the bacterial subgroup had HHV-6B detection in the BAL or plasma, so this group was not further analyzed.

statistical model allowed identification of significantly enriched pathways even when gene-specific differences were modest.

This study also has limitations. There may be residual confounding in the analyses given the observational design, as well as the possibility of misclassification of LRTD categories based on clinical evaluation. Absence of HHV-6B mRNA detection in BALF could be due to variability in mRNA preservation, which could limit interpretation of the ROC analysis; nonetheless, the identified DNA viral load thresholds had clinical significance in the adjusted models. Gene expression analyses were limited by small sample sizes when stratified by HHV-6B detection and clinical disease subgroups precluding adjustment for other variables that could affect the results. Imbalances in specific co-pathogens between groups could have impacted gene expression comparisons. We did not have an independent validation cohort for this unique patient population and sample biorepository, and future studies to validate these findings would be beneficial. Transcriptomic analyses of BAL fluid may provide additional important mechanistic insights. While a randomized controlled prevention or treatment trial would provide the most definitive evidence for HHV-6B as a pulmonary pathogen, such trials would be very difficult, if not impossible, to conduct given the availability of effective antiviral agents and the sample sizes needed. Also, this level of evidence also does not exist for most microorganisms that are widely accepted as pulmonary pathogens after transplantation. Absent this, data from the current study and prior work support HHV-6B as a co-pathogen or primary pathogen in LRTD after allogeneic HCT. Considering commercial availability of similarly sensitive HHV-6 quantitative PCR assays and antiviral treatments (e.g., ganciclovir and foscarnet), there is opportunity to identify and treat affected patients.

In conclusion, we used multiple molecular diagnostic techniques coupled with clinical observations to provide evidence that HHV-6B reactivation in the lungs after allogeneic HCT has immunologically distinct effects on host gene expression and measurable impact on mortality risk, most notably at viral loads ≥2.8 $\log_{10}$ copies/mL in BALF. Higher viral load correlated with detection of HHV-6B gene expression to suggest lytic infection and supported validation of viral load thresholds associated with clinical outcomes. Data from this study fulfill contemporary molecular guidelines for establishing disease causation[55], and recognition of HHV-6B as a pulmonary pathogen has the potential for therapeutic targeting and improvement of outcomes after allogeneic HCT.

## Methods
### Participants
We prospectively enrolled adult allogeneic HCT recipients undergoing a BAL for evaluation of LRTD within 120 days of HCT from Fred Hutchinson Cancer Center (Fred Hutch), City of Hope Comprehensive Cancer Center (COH), and Memorial Sloan Kettering Cancer Center (MSKCC). Data were abstracted from electronic medical records for participant characteristics (including participant sex, which is typically self-reported), microbiologic results, and outcomes for up to 60 days after the BAL date. Gender was not collected. The study was approved by the Fred Hutch Institutional Review Board and participants provided written informed consent. Participants were not compensated.

### LRTD categorization
We defined LRTD as any abnormal radiographic findings with lower respiratory tract symptoms (e.g., cough, shortness of breath, or decreased oxygen saturation). Bacterial, fungal, and viral causes were defined according to consensus guidelines[2,43–45]. Idiopathic pneumonia syndrome (IPS), in which no microbiologically documented infection was detected or suspected, was defined according to the American Thoracic Society guideline[2]. Diffuse alveolar hemorrhage (DAH) was defined as increasingly bloody return from serial lavages or report of "bloody" BALF followed by treatment with corticosteroids[2]. If a microbiologic or non-infectious (i.e., idiopathic pneumonia syndrome [IPS]) cause of LRTD was not determined based on a standardized review of microbiologic, histopathologic, and other diagnostic criteria, we categorized the episode as 'other'. All infectious and non-infectious causes of LRTD were based on review of clinical, microbiologic, histopathologic, and radiographic records. Final determinations were made by author J.A.H.

### Microbiologic testing
The following tests were routinely performed on BALF throughout the study period: gram stain, fungal stain, and acid fast bacilli stain; cytology examination; conventional cultures for bacteria, fungi, mycobacteria, nocardia, and legionella; shell viral centrifugation viral cultures for CMV and/or PCR; and direct fluorescent antibody (DFA) testing and/or PCR for *Pneumocystis jirovecii*. Serum and BALF were routinely tested for *Aspergillus* with the galactomannan index (GMI) using the Bio-Rad Platelia Assay (Hercules, CA)[56]. A GMI ≥ 0.5 with a confirmatory index processed separately on the same sample was considered positive for both BALF and serum. In addition, multiplex quantitative reverse-transcription PCR panel was used to detect up to 12 respiratory viruses including influenza A, influenza B, RSV, PIV types 1 to 4, adenovirus, human metapneumovirus, coronavirus, rhinovirus, and bocavirus[57]. At MSKCC, the BioFire FilmArray Respiratory Panel (Salt Lake City, Utah) was used to detect up to 14 respiratory viruses in addition to *Bordetella pertussis*, *Chlamydophila pneumoniae*, and *Mycoplasma pneumoniae*.

### Procedures, samples, and testing
At all sites, eligible patients routinely underwent a BAL, if clinically appropriate, using standard procedures based on international consensus guidelines[58]. Patients were not routinely tested or treated for HHV-6B in blood or BALF during the study period.

Up to 15 mL of BALF were obtained in real-time during the bronchoscopy and immediately processed. 0.25 mL of fresh BALF were aliquoted and frozen at −80 °C. Using a minimum of 2 mL and up to 10 mL of BALF, a cell pellet was isolated, resuspended in RNAprotect Cell Reagent (QIAGEN, Germantown, MD), and stored at 4 °C for at least 12 h and up to 72 h) before storing at −80 °C. At the time of the BAL procedure, 6 mL of blood were collected in EDTA tubes for plasma isolation and 2.5 mL of blood in PAXgene tubes (QIAGEN, Germantown, MD). PAXgene tubes were stored according to the package insert.

We extracted DNA from fresh BALF aliquots and plasma for HHV-6B testing using real-time quantitative polymerase chain reaction

(qPCR) assays[59]. The assay has a lower limit of detection (LLD) of 10 copies/mL and a lower limit of quantitation (LLQ) of 50 copies/mL; a threshold of ≥50 copies/mL was considered positive. We performed qPCR for the human β-globin gene from BALF to confirm cellularity and the quality of DNA extraction[9,26].

We extracted RNA from BALF cell pellets using the RNAeasy Mini Kit (QIAGEN, Germantown, MD). We extracted RNA from whole blood stored in PAXgene tubes using the MagMAX™ for Stabilized Blood Tubes RNA Isolation Kit and GLOBINclear (Thermo Fisher Scientific, Bothell, WA) to deplete alpha and beta globin mRNA from whole blood samples according to the package inserts. DNase treatment was incorporated into both BAL and blood RNA extractions to reduce contamination by genomic DNA.

We separately tested aliquots of 5 μL of eluted RNA from BAL and blood samples for two HHV-6B mRNA gene transcripts, *U38* (late DNA polymerase gene) and *U90* (immediate early gene), with a laboratory developed reverse-transcription qPCR (RT-qPCR) assay[52]. Synthetically produced RNA transcripts were used for standards. Intra and inter-assay reproducibility were tested, and amplification of house-keeping genes were performed as controls. These targets were previously demonstrated to have high sensitivity and specificity for viral replication[49–53]. We also performed RT-qPCR for GAPDH to confirm the quality of RNA extraction. RNA concentration was assessed using Nanodrop (Thermo Fisher Scientific, Bothell, WA).

Remaining RNA extracted from whole blood samples was processed in the Fred Hutch Genomics & Bioinformatics Shared Resource. We used the TruSeq RNA Exome, previously known as the TruSeq RNA Access kit (Illumina, San Diego, CA), to prepare cDNA libraries. RNA purity was assessed with an Agilent 4200 TapeStation (Agilent Technologies, Santa Clara, CA, USA). Additional library QC, blending of pooled indexed libraries, and cluster optimization was performed using Life Technologies' Invitrogen Qubit® 2.0 Fluorometer (Life Technologies-Invitrogen, Carlsbad, CA, USA). Barcoded RNAseq libraries were pooled (54-plexed) and clustered onto an S1 flow cell. Sequencing was performed using an Illumina NovaSeq 6000 employing a paired-end, 50 base read length (PE50) sequencing strategy. Basecalling and demultiplexing of barcoded reads were performed with Illumina bcl2fastq v2.20. Read quality was summarized using FastQC 0.11.9. Paired reads were aligned to the UCSC GRCh38/hg38 human reference genome using STAR 2.7.7a[60]. Annotations from Gencode v38 were provided with the "--quantMode GeneCounts" option to generate per-gene counts during the alignment process. Only annotations for protein coding genes, polymorphic pseudo-genes, and lncRNAs were considered. The second-strand read counts for each gene were combined into a single gene-count matrix for all samples.

## Statistical analysis

We described the distribution of demographic and clinical characteristics between patients with and without any HHV-6B detection in BALF, as well as the sensitivity and specificity of HHV-6B DNA detection in plasma for predicting HHV-6B DNA detection in BALF. We depicted the distribution of HHV-6B DNA and mRNA detection in BALF by time post-HCT and LRTD category. We generated receiver operating characteristic (ROC) curves to determine the ability of quantitative BALF HHV-6B DNA values to predict HHV-6B mRNA detection.

We used the observed BALF HHV-6B DNA threshold from ROC curves that optimally predicted HHV-6B mRNA detection to define cases of possible HHV-6B pneumonitis. The results from HHV-6B DNA detection were used instead of mRNA detection to improve generalizability for potential clinical applications, as HHV-6 DNA testing is readily available whereas testing for mRNA is not. We generated Kaplan–Meier and cumulative incidence curves to estimate overall mortality and death due to respiratory causes (i.e., 'pulmonary death'), respectively, by 30 and 60 days after BAL in patients with and without possible HHV-6B pneumonitis, irrespective of other causes of LRTD. In the cumulative incidence curves, non-pulmonary deaths were competing events. Univariate and multivariate Cox proportional hazards models were used to compare these outcomes by day 60 after the BAL event, adjusted for potential confounders, including other causes of LRTD. Participant sex was not considered in the study design but was evaluated in univariate models. Two-sided *P* values < 0.05 were considered statistically significant. SAS version 9.3 (SAS institute, Cary, NC) was used for analyses.

For RNA-Seq analyses, we identified subgroups of participants who had a well-documented clinically determined single cause of LRTD (i.e., bacterial, viral, fungal, or IPS). In each diagnostic category, we compared whole-transcriptome expression variability between patients with and without possible HHV-6B pneumonitis using Principal Component Analysis (PCA; Partek Flow [Partek Inc., Chesterfield, MO, USA]). We identified differentially expressed genes using the R (version 4.3) package DESeq2 (version 1.42)[61]. Significant differential gene expression was based on a |log2 (ratio)| ≥ 0.585 (± 1.5-fold) difference and a false discovery rate (FDR) set to 5% to correct for multiple testing[62]. To identify the type of transcriptional programs activated in a given sample, we applied Gene Set Enrichment Analysis (GSEA, version 4.3.1)[63] using 50 Hallmark and 1329 Canonical Pathway gene sets curated from the Molecular Signature Database (MSigDB, version 2023.2; https://www.gsea-msigdb.org/gsea/index.jsp)[64] with a FDR threshold of 5% to denote significantly enriched pathways. Differentially up- or down-regulated gene sets were graphed based on their FDR and normalized enrichment scores using volcano plots.

We performed a priori sample size calculations to estimate target sample sizes and power for detecting differences in overall mortality and gene expression based on preliminary data. We estimated that with a target sample size of approximately 150, we would identify up to 48 participants with any HHV-6B detection in BALF, about half of whom would have detectable HHV-6B mRNA transcripts (n = 24). To test the hypothesis that patients with possible HHV-6B LRTD will have higher mortality than patients without, we estimated that these sample sizes would be able to demonstrate a statistically significant difference if the hazard ratio was at least 2.5–2.6 for effect sizes ranging from 5–20%, with 80% power and a two-sided type I error rate of 0.05. For the subgroup analyses with RNA-Seq, we estimated identifying 10–15 participants per LRTD category who met the described criteria. A power analysis for 2-group comparisons using the RNA-seq Power R package[65] with parameters estimated from a representative RNA-seq dataset demonstrated that the study would be adequately powered (84–95%) to detect meaningful changes in gene expression with ≥1.75–2 n-fold changes assuming a FDR of 5%.

### Reporting summary

Further information on research design is available in the Nature Portfolio Reporting Summary linked to this article.

## Data availability

The RNA-seq data generated in this study meet MINSEQE (Minimum Information About a Next-generation Sequencing Experiment) guidelines and have been deposited in Gene Expression Omnibus under accession code GSE244210 (https://www.ncbi.nlm.nih.gov/geo/). Additional de-identified datasets generated and analyzed for this study are available from the corresponding author (jahill3@fredhutch.org) after publication upon reasonable request, with investigator financial support, and with appropriate documentation of IRB approval and/or data access agreements as applicable. The authors will make best

efforts to provide the requested information within three months of the request.

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

## Acknowledgements

We acknowledge Jeffrey Delrow, Matt Fitzgibbon, Alyssa Dawson, and Pamela Yang for their help with bioinformatics and sample processing. This work was supported by the National Institutes of Health [5K23AI119133 to J.A.H. and K24HL093294 to M.B.] and the American Society for Transplantation and Cellular Therapy/National Marrow Donor Program/Be The Match Foundation [Amy Strelzer Manasevit Research Program] to J.A.H. Additional resources were provided by the National Institutes of Health Fred Hutch/University of Washington/Seattle Children's Cancer Consortium grant [P30 CA015704].

## Author contributions

J.A.H., L.K.V.-V., D.M.Z., A.W., W.M.L., G.-S.C., G.H. and M.B. were responsible for the design of the study and/or interpretation of the data. J.A.H., H.X., W.M.L. and S.A.G. analyzed the data and created the figures. Data were collected by J.A.H., Y.J.L., L.K.V.-V., E.L.C. and S.D. Laboratory work was carried out or supervised by M.-L.H., H.Z. and K.R.J. J.A.H. wrote the first draft. All authors contributed to the writing and revision of the manuscript and approved the final version.

## Competing interests

The authors declare the following financial competing interests. J.A.H.: Consulting for Allovir, Gilead, Karius, Symbio; research support from Allovir, Gilead, Karius, Merck. Y.J.L.: Research support from Karius, AiCuris, Scynexis, and Merck & Co Inc. A.W.: Consulting for Kyorin Pharmaceutical and Vir; research support from Allovir, Ansun Biopharma, Pfizer, Vir/GSK. G.R.H.: Consulting for Generon Corporation, NapaJen Pharma, iTeos Therapeutics, Neoleukin Therapeutics, Commonwealth Serum Laboratories, Cynata Therapeutics; research support from Compass Therapeutics, Syndax Pharmaceuticals, Applied Molecular Transport, Serplus Technology, Heat Biologics, Laevoroc Oncology, iTEOS therapeutics and Genentech. D.M.Z.: Consulting for Allovir. S.S.D.: Consulting for Allovir, Asepticope, Merck, Takeda, Astellas Pharma; research support from Allovir, Ansun Biopharma, Karius, Merck. M.B.: Consulting for Allovir, Symbio, Evrys Bio; research support from Merck. L.K.V., H.X., E.L.C., G.S.C., H.Z., M.L.H., K.R.J., W.L., S.G. and S.J. declare no financial competing interests. All authors have no non-financial interests to declare.
