## [Peer Review File · Nature Communications]

REVIEWER COMMENTS

Reviewer #1 (Remarks to the Author):

The manuscript by Hill et al. reports on a prospective multicenter study of patients undergoing BAL post-HSCT which tests for HHV-6B DNA and RNA in the lung or blood and correlates these results with transcriptomes from blood and disease course. This group has a long standing interest in viruses as agents of occult infection and development of respiratory complications in allo-HSCT. This work largely confirms their past speculations.

The study adds to the field by reporting on these 116 participants and identifying a threshold of HHV6B DNA level which also predicts mRNA expression indicative of lytic infection. This is relevant given that mRNA analyses are not always clinically possible. The study would be strengthened if they could have provided a validation cohort for verification of this result which they don't have.

One of the more interesting novel aspects of this study is the observation that patients with HHV6B high DNA level have distinct host transcriptomics which they highlight as being type I interferon enriched in the IPS patients. While these data are interesting and likely the actual transcriptomic data will be useful to the field, the interpretations are tenuous. For example, interferon is also up in the HHV-6B+ patients with fungal LRTD but the transcriptomes don't segregate, whereas it is not up in the viral LRTD patients, but transcriptomes do segregate. The discussion could better clarify how these results could be interpreted. Also the choice of colors for the pink/red, red/dark red are difficult to distinguish. The transcriptomic signatures and main features in each of these disease groups could use further discussion and analysis, even if in the supplement. (e.g. are there interesting comparisons between the IPS patients and viral LRDT that are unique/informative?)

Overall the data are descriptive, but on balance do support the hypothesis that HHV6B infection/reactivation may drive poor outcomes and development of pulmonary pathology post-HSCT

Reviewer #2 (Remarks to the Author):

Hill et al. conducted a prospective multicenter study including patients undergoing BAL for pneumonia after allo-HCT. The authors quantitated HHV-6 DNA in BAL and paired plasma samples and carried out a transcriptomic analysis in blood from patients testing either positive or negative for HHV-6 DNA in BAL.

The study is of great clinical interest, but I'm not convinced that the data, as presented, clearly implicate HHV-6B as a pulmonary pathogen after allo-HCT, as the authors state. Two major issues be addressed:

1. It is stated that HHV-6B DNA detection above a certain threshold was associated with increased mortality and "pulmonary death" in models adjusted for age, max suppl oxygen and max corticosteroids dose. As the authors indicate, participants with multiple causes of LRTD (58%) had the highest proportion of BALF HHV-6 B detection and higher viral loads; thus, it can be assumed that comparison groups were not matched to co-pathogens presumably involved, which if true may have had an impact on survival.
2. Were comparison groups balanced for CMV DNAemia occurrence at the time of BAL?; How many patients were under (val)ganciclovir treatment at the time of BAL in both groups. This is a crucial issue since ganciclovir has intrinsic activity against HHV-6B.
3. Transcriptomic analyses should have been performed preferentially in BAL specimens to make a stronger case. Again, disbalances across groups in the presence or absence of certain co-pathogens notably makes it difficult to interpret the data.
4. Is it possible that active CMV replication may induce a similar "transcriptomic" response in blood. In this sense, were comparison groups matched for CMV DNAemia?

Reviewer #3 (Remarks to the Author):

In this paper, Hill and colleagues from the Fred Hutchinson Cancer Center, Memorial Sloan Kettering Cancer Center, and the City of Hope National Medical Center pooled clinical data and longitudinally collected specimens to examine the relationship between human herpesvirus 6B (HHV-6B) and lower respiratory tract disease (LRTD) in recent recipients of hematopoietic cell transplants (HCT). They identified a level of HHV-6B DNA detection that was predictive for detection of HHV-6B lytic gene mRNA, and found that participants clinically diagnosed with idiopathic pneumonia in whom HHV-6B DNA was detected in bronchoalveolar fluid had distinct host gene expression signatures and increased risk of death. In the context of modern extensions of Koch's postulates for disease causality, this work provides substantial evidence that HHV-6B is indeed a consequential pathogen in HCT recipients.

The paper addresses an important question. The study design and analysis are logical and sophisticated. The work is clearly presented.

Among other things, this paper is an outstanding example of the importance of high-quality specimen and data collections that connect to diseases of interest, in this instance in recipients of hematopoietic cell transplants.

I have no major criticisms.

Minor points and suggested edits:

1. Explain why detecting transcripts for HHV-6B U38 and U90 indicates active vs. lytic infection.
2. p. 7 top line. "expression" would be better as "detectable expression" or "detection".
3. The GSEA acronym is used on p. 13 and explained on p. 19.
4. It would be helpful to start a new paragraph at the "This study also has limitations" sentence in line 4 of p. 13.
5. In Table 1, add a set with the three hospitals as the variable.
6. The two lower boxes in Fig. 1 are very confusing as presented. I was unable to decipher them with confidence.
7. Fig. 2. Leading digits are covered by the x-axis label in Panels A and B. The on-graph lettering and numbering is too small in panels A and B (including for the colored box legends). "mRNA" rather than "RNA" in the panels A and B on-graph legends.
8. Fig. 3. The "No. of Patients at risk" table is present twice in the figure. Once should suffice. Perhaps more importantly, the HHV-6B-associated deaths happened between day 8 and 30.

Table 1: It seems useful to add a clinical site category to the table (Hutch, etc.).

REVIEWER #1:

The manuscript by Hill et al. reports on a prospective multicenter study of patients undergoing BAL post-HSCT which tests for HHV-6B DNA and RNA in the lung or blood and correlates these results with transcriptomes from blood and disease course. This group has a long standing interest in viruses as agents of occult infection and development of respiratory complications in allo-HSCT. This work largely confirms their past speculations.

The study adds to the field by reporting on these 116 participants and identifying a threshold of HHV6B DNA level which also predicts mRNA expression indicative of lytic infection. This is relevant given that mRNA analyses are not always clinically possible. The study would be strengthened if they could have provided a validation cohort for verification of this result which they don't have.

Response: *We agree with the Reviewer that a validation cohort for this finding would provide additional strength but is unfortunately not available for this unique patient cohort and sample biorepository. We note in the manuscript that the HHV-6B viral load threshold identified by the current study closely aligned with the threshold identified as clinically relevant in a large retrospective study of over 500 similar allogeneic HCT recipients, which adds some degree of validation. This is discussed in the manuscript. We also added the following sentence to the limitations section of the Discussion:*

Page 14: 'We did not have an independent validation cohort for this unique patient population and sample biorepository, and future studies to validate these findings would be beneficial.'

One of the more interesting novel aspects of this study is the observation that patients with HHV6B high DNA level have distinct host transcriptomics which they highlight as being type I interferon enriched in the IPS patients. While these data are interesting and likely the actual transcriptomic data will be useful to the field, the interpretations are tenuous. For example, interferon is also up in the HHV-6B+ patients with fungal LRTD but the transcriptomes don't segregate, whereas it is not up in the viral LRTD patients, but transcriptomes do segregate. The discussion could better clarify how these results could be interpreted.

Response: *Thank you for these important comments. As the Reviewer notes, we highlighted some of the key differences in transcriptional programs in participants with IPS and HHV-6B detection versus without HHV-6B detection; those with HHV-6B detection had enrichment of pathways characteristic of viral infections (e.g., interferon) compared to those without HHV-6B. In this case, the comparator is a group of patients who presumably do not have a concurrent infection and provides the 'cleanest' control group. In the subgroup of patients who had a microbiologically diagnosed viral infection as part of their clinical evaluation, we would not hypothesize to see a difference in interferon pathways (for example), as this is expected to be elevated due to both HHV-6 viral infection as well as the concomitant other viral infection(s). Rather, differences in other unique pathways were observed. Prior studies have also demonstrated the ability of transcriptomics to differentiate even between different viruses or different bacteria, which is similar to what we observe here. We are reassured about the validity of our observations based on detecting similar enrichment of interferon pathways in patients with fungal infection plus HHV-6B detection, as one would expect higher interferon expression in those with concomitant viral infection compared to fungal infection alone.*

The principal components analysis (PCA) was performed using the entire transcriptome for each group comparison, and segregation of samples is based on the global variability in gene expression. In this context, the viral LRTD group (with versus without HHV-6B) had the largest transcriptional signal (as well as the largest number of differentially expressed genes), whereas the fungal group had the weakest signal. Therefore, the separation of samples (HHV-6B vs. no HHV-6B) is more prominent in the viral LRTD group.

We added the following clarifying sentence (underlined) to the results on Page 8:

- *"Multidimensional scaling using principal components analysis (PCA) of the entire transcriptome suggested segregation between patients with viral LRTD who also had HHV-6B detection in BALF or plasma and those without HHV-6B detection; similar segregation by HHV-6B detection was*

evident in the IPS subgroup but not in the fungal subgroup (Figure 4A-4C). This observation indicates that detection of HHV-6B was associated with a stronger transcriptional signal in the viral and IPS subgroups compared to the fungal subgroup.'

We do agree with the Reviewer that our results could be strengthened with a validation cohort given the relatively small data set, but the overall concordance in observations across infection subcategories is reassuring. We highlight this in the limitations of the Discussion and added additional discussion of these findings as detailed below:

- Page 12: 'We did not observe similar enrichment in interferon pathways in the viral infection subgroup, but this was anticipated given that interferon responses are expected to be elevated due to both HHV-6 viral infection as well as non-HHV-6 viral infections, so there was no differential expression relevant to each infection type; rather, differences in other unique pathways were observed.'

Also the choice of colors for the pink/red, red/dark red are difficult to distinguish.

Response: We updated the colors in Figure 4A, 4B, and 4C to improve the visual display of the data as recommended.

The transcriptomic signatures and main features in each of these disease groups could use further discussion and analysis, even if in the supplement. (e.g. are there interesting comparisons between the IPS patients and viral LRDT that are unique/informative?)

Response: We appreciate this important comment from the Reviewer and fully agree that there are many other interesting comparisons within this unique data set. For instance, we do observe distinct transcriptional profiles when comparing across the general categories of viral, fungal, bacterial and IPS causes of lower respiratory tract disease. As a next step, we plan to fully explore these differences in depth for a future publication that broadly evaluates the role of gene expression profiling to facilitate diagnosis of lower respiratory tract disease in HCT patients. However, these observations do not fit in with the scope of the current study, which is focused on the pathogenic role of HHV-6B, but we look forward to presenting these analyses in a separate manuscript in the future.

Overall the data are descriptive, but on balance do support the hypothesis that HHV6B infection/reactivation may drive poor outcomes and development of pulmonary pathology post-HSCT.

Response: We appreciate the Reviewer's thoughtful comments and suggestions.

REVIEWER #2:

Hill et al. conducted a prospective multicenter study including patients undergoing BAL for pneumonia after allo-HCT. The authors quantitated HHV-6 DNA in BAL and paired plasma samples and carried out a transcriptomic analysis in blood from patients testing either positive or negative for HHV-6 DNA in BAL. The study is of great clinical interest, but I'm not convinced that the data, as presented, clearly implicate HHV-6B as a pulmonary pathogen after allo-HCT, as the authors state. Two major issues be addressed:

1. It is stated that HHV-6B DNA detection above a certain threshold was associated with increased mortality and "pulmonary death" in models adjusted for age, max suppl oxygen and max corticosteroids dose. As the authors indicate, participants with multiple causes of LRTD (58%) had the highest proportion of BALF HHV-6 B detection and higher viral loads; thus, it can be assumed that comparison groups were not matched to co-pathogens presumably involved, which if true may have had an impact on survival.

Response: Thank you for raising this question and the opportunity to clarify this important issue. The underlying diagnosis of lower respiratory tract disease (LRTD) was analyzed as a variable in the univariate models as shown in the first two Tables below. There were no significant differences in overall mortality or pulmonary death between underlying diagnoses. As such, this variable did not meet criteria for inclusion in the adjusted models. Nonetheless, to further investigate this, we conducted an updated analysis in an adjusted model including a variable for the clinical diagnoses of LRTD. This demonstrated a similar effect size of HHV-6B on outcomes at the 2.3 log₁₀ copies/ml threshold but an even higher effect size at the 2.8 log₁₀ copies/ml (578 copies/mL) threshold with overall mortality as shown in the updated Tables S4 and new Table S5 below, respectively (relevant analysis in Model 2 in each Table); an adjusted model for pulmonary death was not created due to limited number of events to accommodate this variable.

Overall Mortality Univariate Models for LRTD diagnosis:

Covariates*	Categories	Hazard Ratio (95% CI)	P-values
LRTD cause	Bacterial/Viral/Fungal/Multiple	4.24 (0.58-31.2)	0.156
	IPS	5.36 (0.66-43.6)	0.116
	Other	1 (reference)	
LRTD cause	Bacterial/Viral/Fungal	0.96 (0.44-2.10)	0.919
	Multiple	1 (reference)	
	IPS	1.23 (0.47-3.24)	0.673
	Other	0.23 (0.03-1.80)	0.161

*LRTD variable not further separated due to the limited number of events in the adjusted model to avoid overfitting the model.

Pulmonary Death Univariate Models for LRTD diagnosis:

Covariates*	Categories	Hazard Ratio (95% CI)	P-values
LRTD cause	Bacterial/Viral/Fungal/Multiple	4.24 (0.58-31.2)	0.156
	IPS	5.36 (0.66-43.6)	0.116
	Other	1 (reference)	
LRTD cause	Bacterial/Viral/Fungal	1.13 (0.46-2.81)	0.787
	Multiple	1 (reference)	
	IPS	1.26 (0.40-3.98)	0.689
	Other	0.33 (0.04-2.67)	0.298

*LRTD variable not further separated due to the limited number of events in the adjusted model to avoid overfitting the model.

Table S4. Association of BAL HHV-6B DNA detection ≥ 218 copies/mL with overall mortality and death from respiratory failure by day 60 after the BAL in adjusted Cox models

Variable	Overall mortality		Death from respiratory failure	
	Adjusted HR (95% CI)	P	Adjusted HR (95% CI)	P
Model 1				
Age (years)				
<60	Reference		Reference	
≥ 60	1.50 (0.77-2.94)	0.24	1.26 (0.59-2.67)	0.55
Maximum oxygen use pre-BAL^a				
≤ 2 L/minute	Reference		Reference	
> 2 L/minute	3.27 (1.60-6.71)	0.001	2.76 (1.25-6.09)	0.01
Maximum corticosteroid use pre-BAL^b				
< 1 mg/kg/day	Reference		Reference	
≥ 1 mg/kg/day	1.66 (0.80-3.42)	0.17	0.95 (0.36-2.50)	0.92
HHV-6B DNA in BAL, ≥ 218 copies/mL				
No	Reference		Reference	
Yes	1.81 (0.89-3.67)	0.099	2.35 (1.08-5.11)	0.03
Model 2^c				
LRTD cause				
Bacterial/viral/fungal/multiple	2.35 (0.31-18.0)	0.41	N/A	N/A
IPS	1.84 (0.21-16.4)	0.59	N/A	N/A
Other	Reference	Reference		
Maximum oxygen use pre-BAL^a				
≤ 2 L/minute	Reference		N/A	N/A
> 2 L/minute	3.19 (1.51-6.73)	0.002		
Maximum corticosteroid use pre-BAL^b				
< 1 mg/kg/day	Reference		N/A	N/A
≥ 1 mg/kg/day	1.87 (0.87-4.01)	0.11		
HHV-6B DNA in BAL, ≥ 218 copies/mL				
No	Reference		N/A	N/A
Yes	1.78 (0.87-3.63)	0.11		
Model 3^c				
Age (years)				
<60	Reference			
≥ 60	1.55 (0.79-3.06)	0.20	N/A	N/A
Antiviral use pre-BAL				
No	Reference		Reference	
Yes	2.24 (1.12-4.51)	0.023	2.93 (1.31-6.55)	0.009
Maximum oxygen use pre-BAL^a				
≤ 2 L/minute	Reference		Reference	
> 2 L/minute	2.96 (1.42-6.14)	0.004	2.48 (1.07-5.72)	0.03
Maximum corticosteroid use pre-BAL^b				
< 1 mg/kg/day	Reference		Reference	
≥ 1 mg/kg/day	1.38 (0.66-2.90)	0.39	0.82 (0.28-2.38)	0.71
HHV-6B DNA in BAL, ≥ 218 copies/mL				
No	Reference		Reference	
Yes	1.84 (0.90-3.75)	0.095	2.54 (1.12-5.74)	0.03

CI indicates confidence interval; HHV-6B, human herpesvirus 6B; BAL, bronchoalveolar lavage. 113 participants included in the model. There were 35 events for overall mortality and 27 events for death from respiratory failure (i.e., pulmonary death).

^aWithin 24 hours preceding the BAL.

^bWithin 14 days pre-BAL, based on prednisone-equivalent dosing.

^cIn Model 2, the LRTD variable was not further separated to avoid overfitting due to the limited number of events; inclusion of the LRTD variable in the pulmonary death model could not be accommodated for this reason. In Model 3, age was not incorporated into the pulmonary death model to avoid overfitting.

Table S5. Association of BAL HHV-6B DNA detection ≥ 578 copies/mL with overall mortality and death from respiratory failure by day 60 after the BAL in adjusted Cox models

Variable	Overall mortality		Death from respiratory failure	
	Adjusted HR (95% CI)	P	Adjusted HR (95% CI)	P
Model 1				
Age (years)				
<60	Reference		Reference	
≥ 60	1.41 (0.72-2.77)	0.31	1.12 (0.52-2.46)	0.77
Maximum oxygen use pre-BAL^a				
≤ 2 L/minute	Reference		Reference	
> 2 L/minute	3.65 (1.77-7.54)	< 0.001	3.15 (1.34-7.37)	0.008
Maximum corticosteroid use pre-BAL^b				
< 1 mg/kg/day	Reference		Reference	
≥ 1 mg/kg/day	1.89 (0.90-3.97)	0.09	1.01 (0.39-2.62)	0.99
HHV-6B DNA in BAL, ≥ 578 copies/mL				
No	Reference		Reference	
Yes	2.86 (1.24-6.60)	0.014	2.79 (1.11-7.03)	0.03
Model 2^c				
LRTD cause				
Bacterial/viral/fungal/multiple	2.11 (0.27-16.3)	0.47	N/A	N/A
IPS	1.93 (0.22-17.1)	0.55		
Other	Reference	Reference		
Maximum oxygen use pre-BAL^a				
≤ 2 L/minute	Reference		N/A	N/A
> 2 L/minute	3.09 (1.46-6.57)	0.003		
Maximum corticosteroid use pre-BAL^b				
< 1 mg/kg/day	Reference		N/A	N/A
≥ 1 mg/kg/day	2.08 (0.96-4.50)	0.06		
HHV-6B DNA in BAL, ≥ 578 copies/mL				
No	Reference		N/A	N/A
Yes	3.16 (1.47-6.79)	0.003		
Model 3				
Age (years)				
<60	Reference			
≥ 60	1.66 (0.85-3.26)	0.14	N/A	N/A
Antiviral use pre-BAL				
No	Reference		Reference	
Yes	2.15 (1.08-4.29)	0.03	2.70 (1.23-5.94)	0.01

Maximum oxygen use pre-BAL^a

≤2 L/minute	Reference		Reference	
>2 L/minute	2.91 (1.41-6.03)	0.004	2.41 (1.04-5.54)	0.04

Maximum corticosteroid use pre-BAL^b

<1 mg/kg/day	Reference		Reference	
≥1 mg/kg/day	1.78 (0.83-3.85)	0.14	1.02 (0.35-3.01)	0.97

HHV-6B DNA in BAL, ≥578 copies/mL

No	Reference		Reference	
Yes	3.40 (1.57-7.39)	0.002	3.70 (1.48-9.20)	0.005

CI indicates confidence interval; HHV-6B, human herpesvirus 6B; BAL, bronchoalveolar lavage.

113 participants included in the model. There were 35 events for overall mortality and 27 events for death from respiratory failure (i.e., pulmonary death).

^aWithin 24 hours preceding the BAL.

^bWithin 14 days pre-BAL, based on prednisone-equivalent dosing.

^cIn Model 2, the LRTD variable was not further separated to avoid overfitting due to the limited number of events; inclusion of the LRTD variable in the pulmonary death model could not be accommodated for this reason.

We also note that in a prior retrospective study including a larger number of participants, we demonstrated that the increased risk for mortality associated with HHV-6B detection was observed within each LRTD subcategory (Figure 4 in Hill et al, J Clin Oncol 2019; PMID 31449472; copied below). This also shows that overall mortality was generally similar within each LRTD subcategory (Bacterial pneumonia without HHV-6B detection is the reference group for all the comparisons in the figure below).

We added the following to the Manuscript:

- *Page 7: “There was no significant difference in overall mortality or pulmonary death between underlying LRTD diagnoses, and inclusion of this variable in adjusted models demonstrated a similar effect size of HHV-6B on outcomes at the 2.3 log₁₀ copies/ml threshold but an even higher effect size at the 2.8 log₁₀ copies/ml threshold (Table S4-S5).”*
- *We updated Table S4 to show results from models that incorporated LRTD subgroups and antiviral use using an HHV-6B viral load threshold of 2.3 log₁₀ copies/mL.*

- We added a new Table S5 to show the original model and the 2 additional models incorporating LRTD subgroups and receipt of antivirals in models using an HHV-6B viral load threshold of 2.8 log₁₀ (578) copies/mL (Table S4 uses the 2.3 log₁₀ threshold).
- We slightly adjusted the abstract and concluding paragraph to highlight that we identified relevant viral load thresholds, and based on the updated analyses, the higher threshold of 2.8 log₁₀ copies/mL is probably the most relevant.
- We added Panels C and D to Figure 3, and adjusted the legend accordingly, to depict the incidence curves of overall mortality and pulmonary death when stratified by the higher threshold of 2.8 log₁₀ copies/mL, which showed an even higher risk for mortality than the 2.3 log₁₀ threshold.

2. Were comparison groups balanced for CMV DNAemia occurrence at the time of BAL?; How many patients were under (val)ganciclovir treatment at the time of BAL in both groups. This is a crucial issue since ganciclovir has intrinsic activity against HHV-6B.

Response: We thank the Reviewer for bringing up this important question. Fortunately, all participating sites conducted weekly testing for CMV according to standard practices, and we prospectively captured DNAemia and treatment for CMV and other related dsDNA viruses in this cohort. Overall, the distribution of participants who were receiving antiviral therapy for CMV (and a few for adenovirus) at the time of sample collection was relatively similar across groups as summarized below. There was a suggestion that higher level HHV-6B (above the threshold of 218 copies/mL) was less common in those receiving antivirals, which is not surprising given that HHV-6 reactivation may be suppressed by antiviral therapy (as noted by the Reviewer). This was very similar to findings of our prior retrospective study (Table 1 in Hill et al, J Clin Oncol 2019; PMID 31449472). Although this could impact our findings, one would expect this to bias the HHV-6B negative group to having worse outcomes, as requiring therapy for CMV or adenovirus is associated with lower survival.

To further explore the potential impact of antiviral therapy, we added this variable to the adjusted models. Similar to models incorporated LRTD categories, this demonstrated a similar effect size of HHV-6B on outcomes at the 2.3 log₁₀ copies/ml threshold but an even higher effect size at the 2.8 log₁₀ copies/ml threshold for both overall mortality and pulmonary death as shown in the updated **Table S4** and new **Table S5** above (relevant analysis in Model 3 in each table).

We made the following changes to the manuscript to present these results:

1. We added these data to Table 1 as depicted below.

	Negative HHV-6B DNA in BAL (N=71)	Positive HHV6-B DNA in BAL (any level) (N=42) ^a	Positive HHV6-B DNA in BAL ≥2.3 log ₁₀ copies/mL (N=27) ^a	Total (N=113 ^b)
Antiviral therapy at time of sample collection ^h	24 (34%)	12 (29%)	6 (22%)	36 (32%)

^hGanciclovir, foscarnet, or cidofovir for CMV or adenovirus.

2. We added this sentence to the Results on Page 5: “Of note, a lower proportion of participants with higher HHV-6B viral load were receiving antiviral therapy at the time of sample collection.”
3. We added this sentence to the Results on Page 7: “Additionally, inclusion of a variable for use of antiviral therapy at the time of sample collection resulted in a similar increase in the strength of the association of HHV-6B detection with overall mortality and death from respiratory failure at the higher 2.8 log₁₀ copies/ml threshold (**Tables S4-S5**).”

4. We added the following to the Discussion on Page 12: “There was a suggestion that HHV-6B viral loads were lower in those receiving antivirals targeting related viruses such as CMV and adenovirus, which is not surprising given that these drugs have activity against HHV-6B.²⁵ This was similar to findings of our prior retrospective study.²³ Although this could impact our findings, one would expect this to bias the HHV-6B negative group to having worse outcomes, as requiring treatment for CMV or adenovirus is associated with lower survival.⁵² In adjusted models that accounted for receipt of antiviral therapy, we observed a similar effect size of HHV-6B on outcomes at the 2.3 log₁₀ copies/ml threshold but an even higher effect size at the 2.8 log₁₀ copies/ml threshold for both overall mortality and pulmonary death. Similar findings were demonstrated in models incorporating LRTD subgroups.”

3. Transcriptomic analyses should have been performed preferentially in BAL specimens to make a stronger case. Again, imbalances across groups in the presence or absence of certain co-pathogens notably makes it difficult to interpret the data.

Response: We appreciate this comment and agree that transcriptomic analyses in mRNA isolated from BAL fluid could add additional important mechanistic insights. Unfortunately, the quantity and quality of the RNA isolated from BAL fluid samples was generally poor and not amenable to RNAseq analyses. This was despite great efforts to preserve high quality RNA with bedside collection of samples, prompt processing on site, preservation in RNAProtect, and freezing at -80° Celsius. Nonetheless, we believe the utility of identifying unique gene expression profiles from easily accessible peripheral blood with simple processing (i.e., PAXgene tube collection) is a distinct value demonstrated by this study, as this is a more readily translatable approach for practical clinical applications in the future.

Regarding potential imbalances in the distribution of co-pathogens between groups, we agree that this is an important consideration. To this point, when we compared all patients (regardless of LRTD cause) stratified by HHV-6B detection or not, no unique signal was identified. This was unsurprising given the heterogeneity in expected gene expression profiles driven by different pathogens across the spectrum of viral, fungal, bacterial, and idiopathic causes of LRTD. We addressed the issue of pathogen heterogeneity to the extent possible by performing comparisons stratified by HHV-6B detection within specific diagnostic categories of viral, fungal, or IPS causes of LRTD. Comparisons of HHV-6B detection among patients with a singular cause of pneumonia (e.g, a specific fungal or bacterial species) would require a prohibitively large sample size.

To address these comments, we added the following to the Discussion:

Page 14: “Transcriptomic analyses of BAL fluid may provide additional important mechanistic insights.”

Page 14: “Imbalances in specific co-pathogens between groups could have impacted gene expression comparisons.”

4. Is it possible that active CMV replication may induce a similar “transcriptomic” response in blood. In this sense, were comparison groups matched for CMV DNAemia?

Response: Thank you for bringing up this interesting question, which we agree is important to consider. For the specific subgroups of participants included in the transcriptomic analyses, we summarize the distribution of participants receiving treatment for CMV, including a few treated for adenovirus, below:

IPS Subgroup with N=12 patients in total:

- 4 (33%) participants were receiving antivirals
- None had HHV-6B in the BAL; 1 had HHV-6B in blood only (and was analyzed as an HHV-6B+ case)
- The remaining 3 had no HHV-6B in blood or BAL

Viral Subgroup with N=17 patients in total:

- 8 (47%) participants were receiving antivirals
- 1 had HHV-6B in BAL and blood

- The remaining 7 did not have HHV-6B in BAL or blood

Fungal Subgroup with N=17 patients in total:

- 4 (24%) participants were receiving antivirals
- None had HHV-6B in BAL or blood

Bacterial Subgroup with N=7 patients in total:

- 3 (43%) participants were receiving antivirals
- None had HHV-6B in BAL or blood

In summary, a total of 19 of the 54 patients (35%) in the transcriptomic analyses were receiving antiviral therapy with ganciclovir, foscarnet, or cidofovir at the time of sample collection for this study. Notably, only 1 had HHV-6B in the BALF and the blood, and 1 had HHV-6B in the blood only. Thus, although the use of antivirals was not equally distributed, almost all antiviral treatments are in the 'No HHV-6B' comparator group. This is reassuring, as viremia due to other viruses is unlikely to be impacting the signal for viral infection that was detected in the IPS and fungal groups with HHV-6B, and overall unlikely to impact our conclusions. We considered analyses excluding these individuals, but this would further limit comparisons due to small sample sizes.

We made the following changes to the manuscript:

- 1. We added the details pertaining to treatment for other viruses overall and within each subgroup to Table S7.*
- 2. We added the following sentence to the Results on Page 8: "Of note, 19 of the 54 patients (35%) were receiving antiviral therapy with ganciclovir, foscarnet, or cidofovir for CMV or adenovirus at the time of sample collection for this study. Notably, only two of these patients had HHV-6B in the BALF and/or the blood, so most patients with viremia from another virus were in the 'no HHV-6B' comparator group (Table S7)."*
- 3. We added the following to the Discussion on Page 14: "We also note 19 of the 54 patients (35%) in the transcriptomic analyses were receiving antiviral therapy with ganciclovir, foscarnet, or cidofovir at the time of sample collection for this study. Notably, only 1 had HHV-6B in the BALF and the blood, and 1 had HHV-6B in the blood only. Thus, although the use of antivirals was not equally distributed between comparison groups, almost all events were in the 'No HHV-6B' comparator group. This is reassuring that viremia due to other viruses was unlikely to be driving the observed differences noted in the IPS and fungal groups with HHV-6B detection. The observation that most detection of HHV-6B was in the group not receiving antivirals is concordant with the known activity of these therapies for suppressing HHV-6B reactivation and infection."*

REVIEWER #3:

In this paper, Hill and colleagues from the Fred Hutchinson Cancer Center, Memorial Sloan Kettering Cancer Center, and the City of Hope National Medical Center pooled clinical data and longitudinally collected specimens to examine the relationship between human herpesvirus 6B (HHV-6B) and lower respiratory tract disease (LRTD) in recent recipients of hematopoietic cell transplants (HCT). They identified a level of HHV-6B DNA detection that was predictive for detection of HHV-6B lytic gene mRNA, and found that participants clinically diagnosed with idiopathic pneumonia in whom HHV-6B DNA was detected in bronchoalveolar fluid had distinct host gene expression signatures and increased risk of death. In the context of modern extensions of Koch's postulates for disease causality, this work provides substantial evidence that HHV-6B is indeed a consequential pathogen in HCT recipients.

The paper addresses an important question. The study design and analysis are logical and sophisticated. The work is clearly presented.

Among other things, this paper is an outstanding example of the importance of high-quality specimen and data collections that connect to diseases of interest, in this instance in recipients of hematopoietic cell transplants.

I have no major criticisms.

Minor points and suggested edits:

1. Explain why detecting transcripts for HHV-6B U38 and U90 indicates active vs. lytic infection.

Response: Thank you for this suggestion. We added the underlined parts to the sentences below as indicated:

Page 4: “Establishing a causal link between HHV-6B and LRTD is further hindered by diagnostic methods that do not distinguish between viral shedding (i.e., genomic DNA detection in the absence of viral gene expression indicating latent infection) and lytic infection (i.e., genomic DNA plus viral gene expression indicating viral replication with cytopathic effect).

Page 12: “We used targeted RT-qPCR to test for two HHV-6B-specific transcripts, U38 (a late DNA polymerase gene) and U90 (an immediate early gene), which have high sensitivity and specificity for viral replication in the context of lytic infection and would be unlikely to be detected in the context of an inactive, latent infection. 52–56”

2. p. 7 top line. “expression” would be better as “detectable expression” or “detection”.

Response: We appreciate this suggestion and made this change.

3. The GSEA acronym is used on p. 13 and explained on p. 19.

Response: Thank you for pointing this out. After careful review, we note that GSEA is defined and explained initially on Page 9 before being used on Page 14. It was defined again on Page 21 for the separate Online Methods. As such, we did not make any edits.

Page 9 defines GSEA as follows: “To further examine potential gene expression differences, we performed pathway analysis using Gene Set Enrichment Analysis (GSEA) to define distinct groups of genes that share common biological function, chromosomal location, or regulation in each LRTD category.”

4. It would be helpful to start a new paragraph at the “This study also has limitations” sentence in line 4 of p. 13.

Response: We agree and made this update.

5. In Table 1, add a set with the three hospitals as the variable.

Response: We agree with this suggestion and added the data to Table 1.

6. The two lower boxes in Fig. 1 are very confusing as presented. I was unable to decipher them with confidence.

Response: We agree that the data presented in these boxes was confusing. We updated the consort diagram as depicted below to present the data more clearly. All percentages now use the same denominator of 113 for BAL data and 115 for blood data.

7. Fig. 2. Leading digits are covered by the x-axis label in Panels A and B. The on-graph lettering and numbering is too small in panels A and B (including for the colored box legends). “mRNA” rather than “RNA” in the panels A and B on-graph legends.

Response: We updated these figures as suggested.

8. Fig. 3. The “No. of Patients at risk” table is present twice in the figure. Once should suffice. Perhaps more importantly, the HHV-6B-associated deaths happened between day 8 and 30.

Response: Thank you for this suggestion. We removed this table from below Panel A so that it is now only presented once at the bottom of the figure (below Panel B).

Table 1: It seems useful to add a clinical site category to the table (Hutch, etc.).

Response: We made this update.

REVIEWERS' COMMENTS

Reviewer #2 (Remarks to the Author):

The authors addressed the questions I raise, so I have no further comments.